# Efficacy and Safety of Sofosbuvir/Velpatasvir Plus Ribavirin in Patients with Hepatitis C Virus-Related Decompensated Cirrhosis

**DOI:** 10.3390/v15102026

**Published:** 2023-09-29

**Authors:** Steven Flamm, Eric Lawitz, Brian Borg, Michael Charlton, Charles Landis, K. Rajender Reddy, Mitchell Shiffman, Angel Alsina, Charissa Chang, Natarajan Ravendhran, Candido Hernandez, Christophe Hézode, Stacey Scherbakovsky, Renee-Claude Mercier, Didier Samuel

**Affiliations:** 1Department of Internal Medicine, Division of Digestive Diseases and Nutrition, Rush University Medical Center, Chicago, IL 60612, USA; 2Texas Liver Institute, University of Texas Health San Antonio, San Antonio, TX 78215, USA; 3Southern Therapy and Advanced Research LLC, Jackson, MS 39216, USA; 4Intermountain Medical Center, Murray, UT 84107, USA; 5Division of Gastroenterology and Hepatology, University of Washington, Seattle, WA 98101, USA; 6Department of Medicine, Division of Gastroenterology and Hepatology, University of Pennsylvania, Philadelphia, PA 19104, USA; 7Bon Secours Mercy Health, Liver Institute of Virginia, Richmond, VA 23226, USA; 8Tampa General Medical Group, Tampa, FL 33609, USA; 9Icahn School of Medicine at Mount Sinai, New York, NY 10029, USA; 10Digestive Disease Associates, Catonsville, MD 21228, USA; 11Gilead Sciences, Inc., Foster City, CA 94404, USA; 12Centre Hépatobiliaire, Hôpital Paul-Brousse, Inserm Research Unit 1193, Université Paris-Saclay, 94800 Villejuif, France

**Keywords:** HCV, decompensated cirrhosis, sofosbuvir/velpatasvir, ribavirin, sustained virologic response

## Abstract

A fixed-dose combination of sofosbuvir/velpatasvir (SOF/VEL) plus weight-based ribavirin (RBV) for 12 weeks is recommended for the treatment of patients with hepatitis C virus (HCV)-associated decompensated cirrhosis. However, large global studies, while confirming the effectiveness of SOF/VEL in a broad range of patients, often exclude these patients. This Phase 2, single-arm, open-label study in adult patients with HCV-associated decompensated cirrhosis in France and the USA aimed to provide further data on the safety and efficacy of SOF/VEL plus RBV for 12 weeks in this population. Patients were treated with a fixed-dose combination of SOF 400 mg/VEL 100 mg plus weight-based RBV once daily for 12 weeks. The inclusion criteria were chronic HCV infection (≥6 months), quantifiable HCV RNA at screening, Child–Turcotte–Pugh class B or C cirrhosis, and liver imaging within 6 months of Day 1 to exclude hepatocellular carcinoma. Among 32 patients who initiated treatment, 78.1% achieved sustained virologic response 12 weeks after the end of treatment (SVR12). Failure to achieve SVR12 was due to non-virologic reasons (investigator discretion, *n* = 1; death, *n* = 6). All 25 patients in the per-protocol population achieved SVR12 and all but one achieved sustained virologic response 24 weeks after the end of treatment. Adverse events (AEs) were as expected for a patient population with advanced liver disease. All Grade 3–4 and serious AEs and deaths were deemed unrelated to treatment. In patients with HCV-associated decompensated cirrhosis, SOF/VEL plus RBV achieved high SVR12 rates and was generally well tolerated.

## 1. Introduction

Chronic hepatitis C virus (HCV) infection was estimated to be responsible for approximately 290,000 deaths worldwide in 2019, largely because of liver cirrhosis and hepatocellular carcinoma (HCC) [1]. The development of direct-acting antivirals (DAAs) has revolutionized HCV treatment and prognosis. The pangenotypic regimens sofosbuvir/velpatasvir (SOF/VEL) and glecaprevir/pibrentasvir (GLE/PIB) return high rates of sustained virologic response (SVR) [2,3], improvements in inflammation and fibrosis, and associated improvement in liver function [2].

Guidelines recommend treatment in all patients with HCV without delay, recommending urgent treatment in those with significant fibrosis or cirrhosis [2]. While there are several treatment options for those with compensated cirrhosis (Child–Turcotte–Pugh [CTP] class A), this is not the case for patients with decompensated cirrhosis (CTP class B or C) [2,3].

Treatment regimens comprising a protease inhibitor (PI) are contraindicated in patients with decompensated cirrhosis [2]. Therefore, the PI-free fixed-dose combination of SOF/VEL plus weight-based ribavirin (RBV) for 12 weeks is recommended in management guidelines for the treatment of these patients [2,3]. In the Phase 3, open-label ASTRAL-4 study in patients infected with HCV who had decompensated cirrhosis, sustained virologic response 12 weeks after the end of treatment (SVR12) was achieved by 94% of patients treated with this regimen, with improvements in hepatic function observed across all HCV genotypes [4]. However, of the 87 patients treated with SOF/VEL plus RBV for 12 weeks, only four (5%) had CTP class C cirrhosis at baseline. Although recent clinical trials and real-world studies have demonstrated that 70–100% of patients with CTP class C treated with SOF/VEL can achieve SVR, patient numbers were small and the studies were predominantly in Asia [5,6,7]. Large global studies, while confirming the effectiveness of SOF/VEL in a broad range of patients, have often excluded patients with decompensated cirrhosis [8]. Therefore, the need persists for further data on safety and efficacy of treatment in this population with advanced liver disease.

The aim of this Phase 2 study was to provide further data on the safety and efficacy of SOF/VEL plus RBV for 12 weeks in patients with HCV-associated decompensated cirrhosis, particularly CTP class C, including subjects who had received a liver transplant (LT).

## 2. Methods

### 2.1. Study Design

This Phase 2, single-arm, open-label study in adult patients (males or non-pregnant/non-lactating females aged ≥ 18 years) with chronic hepatitis C and decompensated cirrhosis (CTP class B or C) was conducted in 13 centers in France and the USA from 23 January 2017 to 12 December 2018 (ClinicalTrials.gov NCT02994056). All patients provided written informed consent. The study was conducted according to the Declaration of Helsinki and its amendments and was approved by the appropriate ethics committees.

Patients were eligible for inclusion if they had chronic HCV infection (≥6 months), as documented by either prior medical history or liver biopsy, quantifiable HCV RNA at screening, CTP class B (score 7–9) or C (score 10–15) cirrhosis, and liver imaging within 6 months of Day 1 to exclude HCC. Subjects were non-transplanted or had recurrent HCV following LT. If listed for LT, the projected date of LT was required to be ≥12 weeks after Day 1 of treatment and if post LT, then Day 1 was required to be ≥6 months from the date of the LT. Key exclusion criteria included the following: previous exposure to HCV NS5A inhibitors, portosystemic shunt or variceal bleeding within 6 months of screening, coinfection with HIV or hepatitis B virus, HCC, eGFR_CG_ (estimated glomerular filtration rate by Cockcroft–Gault formula) < 30 mL/min, and platelets < 30,000/μL.

Patients were treated with a fixed-dose combination of SOF 400 mg/VEL 100 mg plus RBV once daily for 12 weeks. RBV was administered at an initial total daily dose of 600 mg, titrated up to a maximum of 1000/1200 mg (1000 mg for patients weighing < 75 kg and 1200 mg for patients weighing ≥ 75 kg) where tolerated. Dose reduction or modification of SOF/VEL was not permitted. Dose modification of RBV was permitted according to the protocol at the discretion of the investigator. Patients were followed for up to 24 weeks after the end of treatment.

### 2.2. Study Assessments

Screening assessments included serum HCV RNA levels and standard laboratory and clinical tests. CTP and the Model for End-stage Liver Disease (MELD) scores were calculated using standard methods [9,10]. Serum HCV RNA was measured using the COBAS^®^ TaqMan^®^ HCV Test v2.0 (F Hoffmann-La Roche Ltd., Basel, Switzerland), with a lower limit of quantification of 15 IU/mL. HCV genotype and subtype were determined using the Siemens VERSANT^®^ HCV Genotype INNO-LiPA 2.0 assay (Siemens Medical Solutions USA, Inc., Malvern, PA, USA). Health-related quality of life (HRQoL) was assessed using the 36-Item Short Form Survey (SF-36), Chronic Liver Disease Questionnaire-HCV (CLDQ-HCV), Functional Assessment of Chronic Illness Therapy-Fatigue (FACIT-F) questionnaire, and the Work Productivity and Activity Impairment: Hepatitis C (WPAI: Hep C) questionnaire [11,12,13,14].

### 2.3. Study Endpoints

The primary efficacy endpoint was the proportion of patients achieving SVR12, defined as HCV RNA < lower limit of quantitation 12 weeks after the end of treatment, in all patients enrolled in the study (intent to treat (ITT)). Secondary efficacy endpoints included the proportion of patients who achieved a sustained virologic response 24 weeks after the end of treatment (SVR24), the number of patients who attained SVR12 after completing 12 weeks of treatment and 12 weeks of follow-up (per protocol [PP], post hoc analysis), and change from baseline in the CTP and MELD scores at the post-end of treatment (EOT) Week 24 in patients who achieved SVR24 and had data available at baseline and Week 24. The effect of treatment on HRQoL was an exploratory endpoint. All subjects who received at least one dose of study drug were included in the safety analysis set. Safety was assessed by incidence of adverse events (AEs), discontinuations due to AEs, and laboratory abnormalities until 30 days after end of treatment.

### 2.4. Statistical Analysis

Point estimates and two-sided 95% exact confidence intervals were calculated based on the Clopper–Pearson method for the proportion of patients achieving SVR12 overall and for the subgroups. Prespecified analyses of changes in CTP and MELD scores from baseline to Week 24 post-EOT were also performed. The post-transplant CTP and MELD scores of patients who had on-study liver transplants were excluded from analysis. Continuous endpoints were summarized using descriptive statistics. Categoric endpoints were summarized by number (*n*) and percentage (%) of subjects who met the endpoint definition. Due to the exploratory nature of this study, no formal power calculations were performed.

## 3. Results

### 3.1. Patient Characteristics

A total of 73 patients were screened, of whom 32 were enrolled in the study (Figure 1). All 32 patients received at least one dose of SOF/VEL plus RBV and were included in both the safety analysis and full analysis sets. Patients were predominantly male, white, and had HCV genotype (GT) 1 infection (Table 1). Over 70% were CTP class C at baseline.

Three patients discontinued study treatment, all for reasons considered unrelated to the study drugs (investigator discretion, *n* = 1; death, *n* = 2). Of the remaining 29 patients who completed treatment, four died prior to the end of the 12-week follow-up period. Therefore, a total of 25 patients completed the full 12-week follow-up period and comprised the PP population (Table 1).

### 3.2. Efficacy

Overall, SVR12 was achieved by 78.1% of patients by ITT analysis (Figure 2). Failure to achieve SVR12 was due to non-virologic reasons (withdrawal of consent, *n* = 1; death, *n* = 6) (Appendix A). All four of the patients who completed treatment but died prior to the end of the 12-week follow-up period had undetectable HCV RNA when last tested while on treatment. All 25 patients in the PP population achieved SVR12. Among patients surviving at Week 24 after the end of treatment, all but one achieved SVR24 (Appendix A). The patient who did not achieve SVR24 due to relapse was infected with HCV GT 3a (Appendix A).

Change in hepatic function was assessed in 19 patients who achieved SVR24, had baseline and Week 24 data available, and did not undergo LT. The CTP score improved in 13/19 patients (68.4%), was stable in 5/19 patients (26.3%), and worsened in 1/19 patients (5.3%) (Figure 3). The MELD score improved in 13/19 patients (68.4%), was stable in 1/19 patients (5.3%), and worsened in 5/19 patients (26.3%) (Figure 4). Overall, the majority of patients who achieved SVR24 saw an improvement in hepatic function.

### 3.3. Impact on HRQoL

During the treatment period, improvements were seen in the SF-36 mental component score, the overall CLDQ-HCV score, and the WPAI: percent activity impairment due to HCV scale. However, the mean changes to the SF-36, CLDQ-HCV, FACIT-F, and WPAI scores between baseline and EOT were not statistically significant.

The mean scores for most scales improved from EOT to Week 12 post-EOT. However, the only statistically significant change was an improvement in the WPAI: percent activity impairment due to HCV scale between EOT and Week 12 post-EOT (*p* = 0.003).

Overall, results from the SF-36, CLDQ-HCV, FACIT-F, and WPAI questionnaires indicated that no statistically significant worsening in HRQoL was observed between baseline and EOT.

Patients who had an improved CTP score between baseline and Week 12 post-EOT showed an improvement in the mean SF-36 mental component score, CLDQ-HCV score, and FACIT-F score, but a worsening of the SF-36 physical component score. Patients who did not have an improved CTP score between baseline and Week 12 post-EOT showed a worsening in all HRQoL mean scores (Appendix A).

### 3.4. Safety

Observed AEs were consistent with expectations for a patient population with advanced liver disease (Table 2). All Grade 3–4 and serious AEs and deaths were deemed unrelated to treatment. The most reported Grade 3 or 4 laboratory abnormalities were consistent with the known safety profile of RBV. Two patients discontinued treatment due to AEs, both for reasons considered unrelated to treatment. In all, three patients underwent liver transplant during the study period at Days 13, 34, and 77 after the end of treatment. All achieved SVR12 and SVR24.

Eight patients died during the study period. Two deaths occurred prior to treatment completion, four during follow-up, and two after the Week 12 follow-up visit. The causes of death were liver failure (*n* = 2), sepsis (*n* = 2), cardiac arrest (*n* = 1), colitis (*n* = 1), acute pancreatitis (*n* = 1), and variceal hemorrhage (*n* = 1). All deaths were deemed unrelated to treatment.

The mean daily dose of RBV was 611 ± 82.2 mg, with a mean weight-based amount of 7.4 ± 2.48 mg/kg for a mean of 9.5 ± 4.5 weeks or 66 ± 31.3 days. Overall, eight patients (25.0%) discontinued RBV prematurely with a further eight interrupting (*n* = 4) or reducing (*n* = 4) the dose for ≥3 consecutive days.

## 4. Discussion

In patients with chronic HCV and decompensated cirrhosis, SOF/VEL plus RBV for 12 weeks achieved high rates of SVR in both CTP B and CTP C patients. While SVR12 by ITT analysis was lower than that reported by others [4,5,6]; this was because almost one-quarter of enrolled patients did not complete the 12-week post-EOT follow-up period, predominantly due to deaths unrelated to treatment in the CTP C group. Furthermore, all four of the patients who completed treatment but died prior to the end of the 12-week follow-up period had undetectable HCV RNA when last tested on treatment. All patients who completed treatment and follow-up achieved SVR12, and only one patient with GT 3 infection experienced virologic failure by Week 24 post-EOT. The efficacy in patients with CTP class B cirrhosis was in line with that observed in patients without cirrhosis (99%), or with compensated cirrhosis (95%) treated with SOF/VEL in the ASTRAL-2 and ASTRAL-3 trials, respectively [15]. These results are encouraging as treatment regimens comprising a PI are contraindicated in patients with decompensated cirrhosis, so SOF/VEL is the only approved DAA that can be administered to these patients [2].

While we did not evaluate the effect of RBV on treatment outcomes in our study, there is some evidence that the addition of RBV to SOF-based regimens when treating patients with HCV and decompensated cirrhosis may not be beneficial. A recent meta-analysis reported similar SVR rates in patients with HCV and decompensated cirrhosis treated with SOF-based regimens with or without RBV [16]. The same meta-analysis also found that in patients treated with SOF/VEL, the frequency of AEs was significantly higher in those with RBV than in those without RBV (91% vs. 50%; *p* = 0.008) [16]. In our study, anemia was the most commonly reported AE leading to the modification or interruption of RBV, which is consistent with the known toxicity profile of RBV [17,18,19]. As safety is a key limitation of treatment in patients with HCV and decompensated cirrhosis, further study is needed to confirm whether the addition of RBV to SOF/VEL is beneficial in these patients.

In the current study, improvements in liver function were seen in over one-half of patients who achieved SVR24. This is consistent with the findings of the ASTRAL-4 study, which reported early improvements in CTP class and MELD score in patients with CTP class B cirrhosis who achieved SVR12 [4]. Around one-half of patients in the ASTRAL-4 study showed an improvement in CTP score from baseline, while 11% showed a worsening [4]. In the same study, the majority of patients showed improved MELD scores, most markedly those with baseline scores of 15 or more [4]. Similar results have been reported for CTP B patients in other studies [5,6]. Although improvements in CTP class and MELD score have been reported in CTP C patients who achieve SVR12, the data for patients with CTP C decompensated cirrhosis from these trials are more limited [5,6]. Given the dearth of data in these patients, it is reassuring to note that in our study, over one-half of patients with available data who were CTP class C at baseline and achieved SVR24 had improved to CTP class B by Week 24 post-EOT.

Maintenance or improvements in CTP class in patients with decompensated cirrhosis treated with SOF-based regimens have been shown up to 96 weeks post-EOT [20]. An analysis of patients with decompensated cirrhosis included in the HCV-TARGET cohort who achieved SVR after DAA treatment showed that liver function was relatively stable over 4 years, although with only marginal improvements in MELD score [21]. The relevance of improvements in CTP class and MELD score to long-term clinical outcomes is unclear. In addition, there is some debate around whether to administer treatment before or after liver transplant [22,23]. While DAA treatment in some cirrhotic patients can improve liver function sufficiently to allow delisting [24], in others, improvement may just deprioritize them for transplantation without significantly improving quality of life, so-called ‘MELD purgatory’. In these patients, treatment post LT may be more beneficial. Treatment before liver transplant may also reduce the number of appropriate donors for an individual patient, as it can prevent them receiving a liver from an HCV-positive donor. One US study that estimated the impact of providing HCV treatment before or after liver transplantation in decompensated patients determined that the MELD score threshold below which patients should be treated before liver transplant is between 23 and 27, depending on region, as defined by the United Network for Organ Sharing [25]. This study found that treating patients with MELD scores below this threshold may be beneficial, whereas treating patients above this threshold may be associated with decreased life expectancy [25]. The International Liver Transplant Society recommends that patients with a MELD score < 20 and/or CTP class B should be treated before liver transplant [26]. It was not possible to define this threshold in our study, as only four patients who achieved SVR24 (without undergoing a liver transplant) had a baseline MELD score ≥ 20. Our study data are particularly valuable, as the majority of patients that were included meet the criteria for treatment pre liver transplant. Furthermore, liver transplants are still not readily available in some countries [27,28].

Some studies have also indicated that attainment of SVR with DAAs significantly reduces, although does not eliminate, the risk of developing HCC in patients with HCV-associated cirrhosis [29,30]. However, a recent international multicenter cohort study showed that, while achievement of SVR after DAA treatment was associated with improved event-free survival (liver transplant or death) in patients with CTP A cirrhosis, no such association was seen for those with decompensated cirrhosis, despite improvements in MELD score [31]. In our study, while the majority of patients achieved SVR saw improvements in CTP class and MELD score, longer-term follow-up would be required to confirm whether SVR was associated with a reduced risk of clinical disease progression. Overall, these findings emphasize that the close monitoring of patients with decompensated cirrhosis is required, even after the achievement of SVR.

In our study, overall results from the HRQoL questionnaires indicated that no statistically significant worsening in HRQoL was observed between baseline and EOT. Other studies have found that patients with decompensated cirrhosis have significantly worse HRQoL than compensated or non-cirrhotic patients [32,33]. While there is potential to improve patient HRQoL through effective treatment, in the relatively short period of observation in this study, a substantial improvement in patients with already poor HRQoL cannot be expected. These results should be interpreted with caution, as multiple endpoints were tested, and the study was not powered to test these exploratory endpoints.

The relatively small number of patients limits assessment of the impact of characteristics such as genotype on response. However, compared with other studies of SOF/VEL plus RBV in decompensated cirrhosis, a relatively high proportion of patients with CTP C were included. These data are particularly valuable in a population that still has few treatment options and, to our knowledge, are some of the first data outside of Asian populations.

In conclusion, in patients with chronic HCV infection and either CTP B or CTP C cirrhosis, SOF/VEL plus RBV achieved high cure rates, with a low rate of virologic failure to Week 24 post-EOT. Clinically relevant improvements in liver function, as demonstrated by improvements in CTP class and MELD score, were observed. Treatment was generally well tolerated, with observed AEs consistent with expectations for a patient population with advanced liver disease.

## Figures and Tables

**Figure 1 viruses-15-02026-f001:**
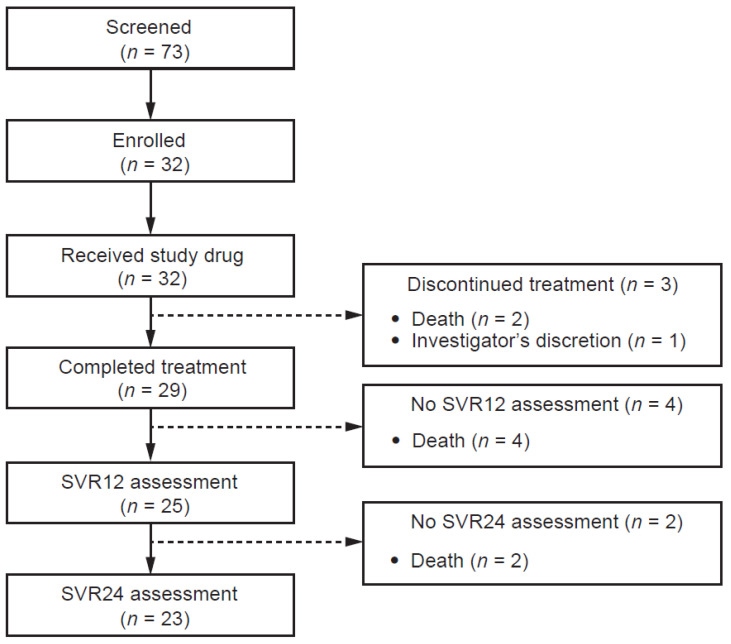
Patient characteristics (screened subjects). SVR12, sustained virologic response 12 weeks after the end of treatment; SVR24, sustained virologic response 24 weeks after the end of treatment.

**Figure 2 viruses-15-02026-f002:**
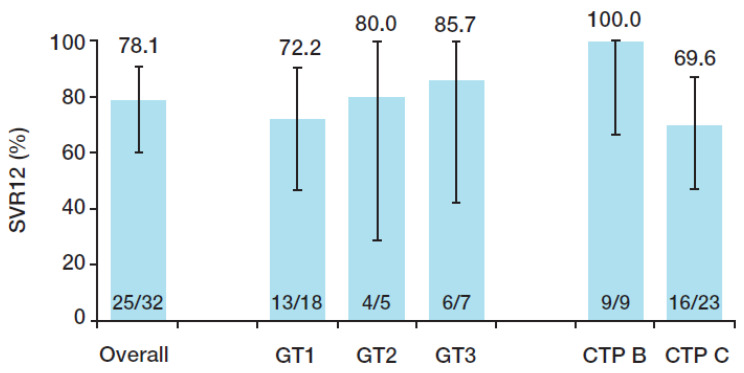
Percentage of patients achieving SVR12 overall and by subgroup (ITT analysis). The rates of SVR12 overall and by subgroup are shown. The viral load of two subjects included in the overall population was too low to assess genotype. The I bars represent 95% confidence intervals. CTP, Child–Turcotte–Pugh; ITT, intent-to-treat; GT, genotype; SVR12, sustained virologic response 12 weeks after the end of treatment.

**Figure 3 viruses-15-02026-f003:**
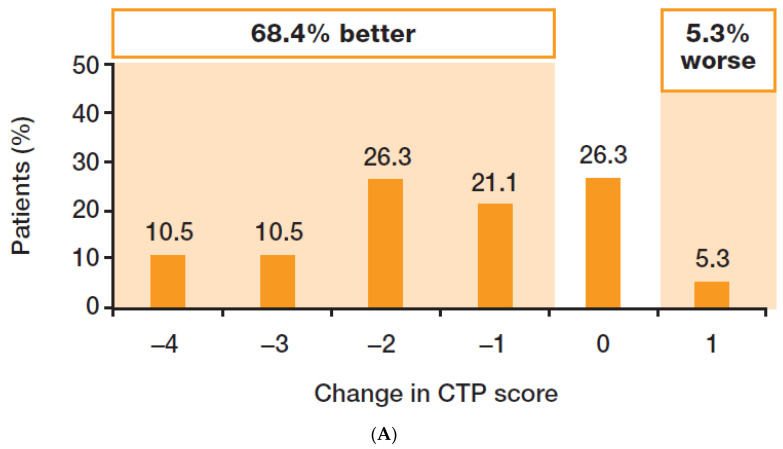
Change in CTP score (baseline to Week 24 in patients with SVR24 and liver function data available, *n* = 19). The changes in CTP score are shown from baseline to Week 24 in patients with SVR24 who had baseline and Week 24 data available and did not undergo liver transplant. CTP, Child–Turcotte–Pugh; SVR24, sustained virologic response 24 weeks after the end of treatment.

**Figure 4 viruses-15-02026-f004:**
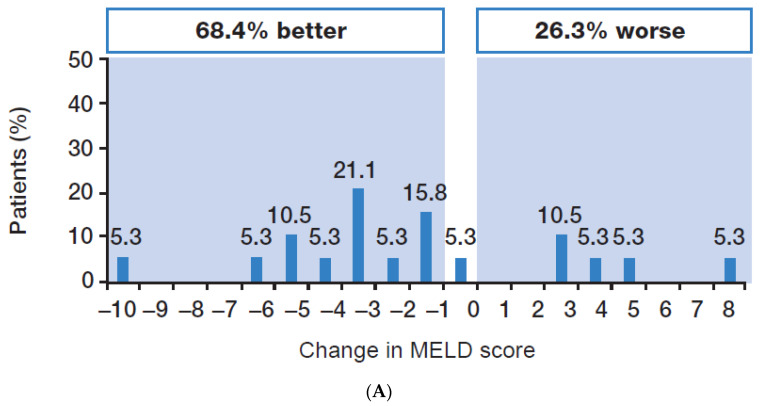
Change in MELD score (baseline to Week 24 in patients with SVR24 and liver function data available, *n* = 19). The changes in MELD score from baseline to Week 24 are shown in patients with SVR24 who had baseline and Week 24 data available and did not undergo liver transplant. MELD, Model for End-stage Liver Disease; SVR24, sustained virologic response 24 weeks after the end of treatment.

**Table 1 viruses-15-02026-t001:** Baseline demographics and disease characteristics.

	SOF/VEL + RBV 12 Weeks	
	ITT Analysis *n* = 32	PP Analysis *n* = 25
Mean age, years (range)	55 (39–77)	54 (39–67)
Male, *n* (%)	26 (81.3)	19 (76.0)
Race, *n* (%)		
White	18 (56.3)	14 (56.0)
Black	6 (18.8)	4 (16.0)
Other/not stated	8 (25.0)	7 (28.0)
Country of enrollment, *n* (%)		
USA	26 (81.3)	19 (76.0)
France	6 (18.8)	6 (24.0)
HCV GT, *n* (%)		
1a 1b	16 (50.0) 2 (6.3)	11 (44.0) 2 (8.0)
2	5 (15.6)	4 (16.0)
3	7 (21.9)	6 (24.0)
Indeterminate *	2 (6.3)	2 (8.0)
CTP class, *n* (%)		
B (7–9)	9 (28.1)	9 (36.0)
7	0 (0.0)	0 (0.0)
8 9	2 (6.3) 7 (21.9)	2 (8.0) 7 (28.0)
C (10–15)	23 (71.9)	16 (64.0)
10	11 (34.4)	9 (36.0)
11	6 (18.8)	3 (12.0)
12	4 (12.5)	3 (12.0)
13	2 (6.3)	1 (4.0)
14	0 (0.0)	0 (0.0)
15	0 (0.0)	0 (0.0)
MELD score, *n* (%)		
10–15	13 (40.6)	10 (40.0)
16–20	17 (53.1)	14 (56.0)
21–25	2 (6.3)	1 (4.0)
Ascites, *n* (%)		
None	3 (9.4)	3 (12.0)
Mild/moderate	20 (62.5)	17 (68.0)
Severe	9 (28.1)	5 (20.0)
Encephalopathy, *n* (%)		
None	5 (15.6)	5 (20.0)
Medication-controlled	27 (84.4)	20 (80.0)
HCV RNA, log_10_ IU/mL mean (SD)	5.2 (1.2)	5.1 (1.2)
HCV RNA ≥ 800,000 IU/mL, *n* (%)	9 (28.1)	7 (28.0)
HCV treatment experienced, *n* (%) ^†^	4 (12.5)	4 (16.0)
eGFR_CG_, mL/min, mean (SD)	113.3 (41.2)	111.0 (41.8)
Platelets, ×10^3^/µL, mean (SD)	89.1 (30.8)	92.6 (30.8)
Albumin, g/dL, mean (SD)	2.7 (0.5)	2.8 (0.5)
INR, mean (SD)	1.5 (0.3)	1.5 (0.3)
Hemoglobin, g/dL, mean (SD)	12.0 (1.3)	12.0 (1.3)
Lymphocytes, ×10^3^/µL (SD)	1.6 (0.9)	1.7 (0.9)
Bilirubin, mg/dL, mean (SD)	3.4 (1.5)	3.4 (1.5)

* Viral load too low to assess. ^†^ Treatment-experienced subjects must have completed their most recent HCV treatment at least 8 weeks prior to screening. CTP, Child–Turcotte–Pugh; eGFR_CG_, estimated glomerular filtration rate by Cockcroft–Gault formula; GT, genotype; INR, international normalized ratio; ITT, intent to treat; MELD, Model for End-stage Liver Disease; PP, per protocol; RBV, ribavirin; SD, standard deviation; SOF/VEL, sofosbuvir/velpatasvir.

**Table 2 viruses-15-02026-t002:** Adverse events.

Event	SOF/VEL + RBV 12 Weeks *n* = 32
Any AE, *n* (%)	31 (96.9)
AEs seen in ≥5% of patients, *n* (%)	
Anemia *	8 (25.0)
Nausea	8 (25.0)
Asthenia	6 (18.8)
Vomiting	5 (15.6)
Grade 3–4 AEs, *n* (%)	11 (34.4) ^†^
Serious AEs, *n* (%)	17 (53.1) ^†^
SAEs seen in ≥5% of patients	
Hepatic hydrothorax	2 (6.3)
Hepatocellular carcinoma	3 (9.4)
Sepsis	2 (6.3)
AE leading to discontinuation of all study drugs, *n* (%)	2 (6.3) ^†‡^
Deaths, *n* (%)	8 (25.0) ^†^
Laboratory abnormalities seen in >1 patient, *n* (%)	
Grade 3	11 (34.4)
Low lymphocytes (350–<500/mm^3^)	4 (12.5)
Low platelets (25,000–<50,000/mm^3^)	4 (12.5)
Hyperbilirubinemia (>2.5–5.0 × ULN)	6 (18.8)
Hyperuricemia (>12.0–15.0 mg/dL)	5 (15.6)
Grade 4	7 (21.9)
Hyperbilirubinemia (>5.0 × ULN)	5 (15.6)

* Diagnosed at the discretion of the investigator; ^†^ Unrelated to treatment; ^‡^ Grade 4 sepsis and Grade 4 respiratory distress followed by probable sepsis, both resulting in death. AE, adverse event; RBV, ribavirin; ULN, upper limit of normal; SAE, serious adverse event; SOF/VEL, sofosbuvir/velpatasvir.

## Data Availability

Data are contained within the article.

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
