# Peer review of "Efficacy and Safety of Sofosbuvir/Velpatasvir Plus Ribavirin in Patients with Hepatitis C Virus-Related Decompensated Cirrhosis"

_viruses, 2023, doi:10.3390/v15102026_

Round 1

Reviewer 1 Report

In the study "Efficacy and safety of sofosbuvir/velpatasvir plus ribavirin in patients with hepatitis C virus-related decompensated cirrhosis", Flamm et al. describe the results of an open-label, phase 2 trial assessing safety and efficacy of SOF/VEL + RBV in patients Child Pugh Classes B and C cirrhosis. This study and its findings are significant due to the high morbidity and mortality in this patient population and difficulties with managing the toxicity profile of Ribavirin.  The manuscript is well-written and easy to follow with concise description of results and a thoughtful discussion.  A major strength of the study is the number of patients with Child Pugh Class C and GT3 infection. There are some clarifications and improvements needed in the methods and results to enhance the manuscript.

Methods: 

1. Were all patients treated in a liver transplant center and were all treating clinicians gasteroenterologists/hepatologists? If not, this should also be included in the discussion. There is hesitancy to treat patients with decompensated cirrhosis if the provider does not practice hepatology and/or patients do not have access to a liver transplant clinic.

2. Section 2.2: please provide references validating the use of the health-related quality of life assessments

3. Section 2.3: Recommend not saying "post treatment" and use "end of treatment" instead t. Post-treatment can be mis-interpreted to mean the number of weeks after Day 0 of treatment. 

4. Please clarify why patients with prior NS5A exposure were excluded. Per AASLD guidelines, these patients can still be treated, with prolonged treatment of 24 weeks, and EASL guidelines state these patients can be given SOF/VEL + RBV for 12 weeks. Not including these patients is a limitation to the study which should be included in the discussion. 

Results:

1. Table 1: include number of LT patients. Also indicate if any of the previously treated patients had been treated with an NS5B inhibitor. 

2.  Figure 3A: make y axis either absolute number of patients or % and remove line below with the absolute number 

3. Figure 4A is mis-labeled and should be "change in MELD score". Same comment as Figure 3A, remove the absolute number of patients from the row below the y axis and make the y axis either number of patients or %. 

4. Figures 3B and 4B seem redundant compared to Figures 3A and 4A, respectively. Consider presenting only Figures 3A and 4A; these figures convey the improvement in CTP and MELD scores concisely. 

5. Were any of the deaths or SVR failures in patients with liver transplants? If so, immunosuppression may play a role in the deaths (especially with the 3 who died from sepsis) or failures. 

Reviewer 2 Report

Flamm et al. reports the efficacy and safety of sofosbuvir/velpatasvir/ribavirin in chronic hepatitis C patients with decompensated cirrhosis. Overall, the study provides valuable clinical data and supports the use of sofosbuvir/velpatasvir in this subpopulation. Therefore, the study is important. The paper is well written and data are clearly presented. One minor comment is that since the trial was completed in 2018, it might be more informative to include any follow-up data if available.
